

# Compensatory growth and understory soil stoichiometric features of *Hippophae rhamnoides* at different stubble heights

Xin Wang[1], Yuefeng Guo[1], Wei Qi[2], Li Zhen[3], Yunfeng Yao[1] and Fucang Qin[1]

[1] College of Desert Control Science and Engineering, Inner Mongolia Agricultural University, Hohhot, Inner Mongolia, China
[2] Inner Mongolia Autonomous Region Water Conservancy Development Center, Inner Mongolia Autonomous Region Water Conservancy Development Center, Hohot, Inner Mongolia, China
[3] College of Forestry, Northeast Forestry University, College of forestry, Northeast Forestry University, Harbin, Heilongjiang, China

Corresponding author
Yuefeng Guo,
wxdiuam@emails.imau.edu.cn

## ABSTRACT

**Background**. This study was aimed to explore the compensatory growth ability and influence mechanism of *Hippophae rhamnoides* at the decaying phase in feldspathic sandstone areas of Ordos, and clarify the stubble height when the compensatory growth ability of *H. rhamnoides* was the strongest.

**Methods**. The *H. rhamnoides* forests in the decaying phase from an exposed feldspathic sandstone zone of Ordos were chosen. The compensatory growth ability of *H. rhamnoides* at stubble height of 0 cm ($S_1$), 10 cm ($S_2$), 15 cm ($S_3$), 20 cm ($S_4$) and control (CK) was investigated with *H. rhamnoides* forests at the decaying stage in the exposed feldspathic sandstone areas of Ordos. Relationships of compensatory growth ability of *H. rhamnoides* and understory soil properties with understory soil stoichiometric features as well as the response mechanism to stubble height were explored.

**Results**. (1) Overcompensatory growth of *H. rhamnoides* in feldspathic sandstone areas occurred at all stubble heights. Especially, the plant height compensation index (1.45) and biomass compensation index (1.25) at the stubble height of 15 cm were both larger compared with other stubbling treatments. These results indicate the stubble height of 15 cm can well promote the growth of the ground part of *H. rhamnoides*. (2) All stubble heights significantly affected the contents and eco-stoichiometric ratios of soil organic carbon (SOC), total nitrogen (TN), total phosphorus (TP) in understory soils, but the influence rules differed. SOC, TN, and TP contents at all stubble heights were larger than those of the control, and maximized at the stubble height of 15 cm. The carbon(C): phosphorus(P) ratio, and nitrogen (N):(P) ratio after stubbling treatments were all lower compared with the control, and minimized to 19.52 and 1.84 respectively at the stubble height of 15 cm. (3) The understory C:N:P stoichiometric ratio of *H. rhamnoides* in feldspathic sandstone areas is jointly affected by compensatory growth, stubble height, and soil physicochemical properties. The total explanation rate determined from RDA is 93.1%. The understory soil eco-stoichiometric ratio of *H. rhamnoides* is mainly affected by soil moisture content (contribution of 87.6%) and total porosity (7.9%), indicating soil moisture content is the most influential factor. The findings will offer some new clues for eco-construction and theoretically underlie soil-water loss administration.

# INTRODUCTION

Feldspathic sandstone is widely distributed in Jungar Banner, Ordos, China. This loose stratum mainly comprises of terrigenous clastic rocks including red-white interphase sandstone, siltstone, and mudstone. Because of low-degree consolidation between sand grains, feldspathic sandstone cannot be easily destroyed under normal conditions, but becomes mud upon watering and turns into sand upon winding (*Guo et al., 2021*). Feldspathic sandstone is locally named "Pi sandstone" (Pi or pi shuang is the folk name of arsenic in Chinese), which implies feldspathic sandstone is as harmful as arsenic to soil and water conservation (*Wang et al., 2007*). Hence, massive water and soil loss occurs easily in feldspathic sandstone areas, which carries off numerous nutrients from soils and leads to soil fertility decline and soil leanness, inducing land desertification and severe soil erosion. Hence, experimental study in this region is beneficial for ecosystem restoration there.

Stumping or cradling is a major measure for the renewal and rejuvenation of plants. Specifically, the branches above the root collar are cut off at a certain height, which will enhance the sprouting of seedlings and drive the seedlings to grow thick and strong branches (*Zhong, 2020*). The active response of plants to stumping or cradling is called compensatory growth (*Zhao, Chen & Lin, 2008*). Depending on whether the accumulative ground biomass of cradled plants is more than, equal to, or less than that of the non-cradled plants (*Zhao, Chen & Lin, 2008*; *Liu et al., 2011*), plant compensatory growth is divided into 3 types: overcompensation, equivalent compensation and undercompensation (*Liu et al., 2011*). Appropriate cradling can promote the tillering and photosynthesis of cereal grass plants, and accelerate growth recovery, leading to equivalent compensation or over-compensatory growth (*Belsky, 1986*). Excessive stubble (*Burns, Chamblee & Giesbrecht, 2002*) and low stubble (*Zhao, Chen & Lin, 2008*; *Zhong & Bao, 1999*) both are unfavorable for pasture production. Hence, stubble height is a key parameter in management of vegetation stumping. *Cutforth & McConkey (1997)* studied spring wheat for 4 years in semiarid Canada grassland, and found the yield of spring wheat was raised by 13% after high stubbling and by 4% after short stubbling. Different from the finding of *Cutforth & McConkey (1997)* and *Zhao, Chen & Lin (2008)* found high stubble promoted the revegetation of *Leymus chinensis*, but short stubbles lowered revegetation. This may be caused by differences in vegetation types, adaptability to stubble height, or soil types between the two study regions. The existing experimental findings about the suitable stubble height are inconsistent among different studies and between experimental research and practical application, and many factors may influence the results, including experimental design, vegetation type, site conditions, soil type, and stumping time. Given the special soil type—feldspathic sandstone in Ordos—it is necessary to further study the effects of stubble height on vegetation, which will promote the sustainable stubble height development in this region. *Shao et al. (2008)* state that soil physicochemical properties contribute to vegetation recovery. As reported,

removal of ground biomass by mowing directly affected the vegetation, topsoil litters, and soil carbon-nitrogen sinks (*Jaramillo & Detling, 1988*; *Ziter & MacDougall, 2013*). Multiple linear regression from *Shahrudin et al. (2014)* showed that the ground biomass of vegetation can best explain the variations of soil organic matter accumulation rate. *Ziter & MacDougall (2013)* found stumping can promote the soil carbon fixation stocks of vegetation through the roots, which may be a response to the compensatory growth lost on the leaf surface. Stumping can indirectly increase soil temperature and humidity and thereby improve nutrient contents or nutrient absorption and alter the relative distribution of nutrients (*Jaramillo & Detling, 1988*; *Han et al., 2014*). *Yu (2016)* studied the effects of stumping on the physiological properties of *Caragana microphylla* and the physicochemical properties of soils, and found soil C, N and P contents at the end of growing seasons were higher and soil potential of hydrogen (pH) and bulk density were lower than at the early stage. *Brejda et al. (2000a)* and *Brejda et al. (2000b)* thought organic carbon was the most potential indicator of soil quality, and nitrogen and phosphorus were usually the restricting nutrients of plant growth (*Güsewell, 2004*; *Aerts & Chapin, 1999*). The findings are different among studies, owing to differences in the study areas, vegetation ad soil types. The feldspathic sandstone in Ordos contains low soil nutrients, and there is no discussion about how the understory carbon, nitrogen and phosphorus concentrations and their stoichiometric ratios in *H. rhamnoides* forests respond to stubble heights. For these reasons, we are restricted from predicting the feedback ability of vegetation growth to stubble heights in this region, which calls for further research.

*Hippophae rhamnoides* is a deciduous shrub or small arbor belonging to *Hippophae*, Elaeagnaceae Juss (*Chen, 2017*). The well-developed lateral roots with strong sprouting ability of *H. rhamnoides*, which can strongly occupy the underground space and consolidate soils and prevent soil-water loss. Thus, *H. rhamnoides* is extensively planted in the feldspathic sandstone areas of Ordos, Inner Mongolia. However, owing to the special soil properties, specific geological conditions and drought in feldspathic sandstone areas, the *H. rhamnoides* forests planted there will decay in both growth rate and productivity at the age of 10 years (*Yang et al., 2005*; *Hao et al., 2005*). Under such a background, we tried to probe into the effects of stubble heights on the compensatory growth ability and understory soil physicochemical properties of *H. rhamnoides* in feldspathic sandstone areas, and aimed to find out whether compensatory growth of *H. rhamnoides* will occur after stumping and whether the understory soil physicochemical properties can be improved after stumping. We aim to uncover the regulatory mechanism between the compensatory growth ability and understory soil physicochemical properties of *H. rhamnoides* at different stubble heights in feldspathic sandstone areas. This study will enrich the research contents on the adaption of *H. rhamnoides* to feldspathic sandstone areas and can accelerate administration of soil-water loss. Our findings are valuable for resource utilization in feldspathic sandstone areas.

## MATERIALS & METHODS

### The study area

The study area is located in the Geqiu groove watershed (39°42′–39°50′N, 110°25′–110°48′E) of Nuanshui Village, Jungar Banner, Ordos, Inner Mongolia (Fig. 1). This watershed has fluctuating terrains and numerous gullies, with fluctuating girders, but suffers intense soil erosion and severe soil-water loss. With an average altitude of 1044 m, this area enjoys a typical moderate-temperature semiarid continental monsoon climate, with average sunshine duration of 3000 h and frost-free period of 148 days. The annual average precipitation is about 400 mm, which is concentrated in July and August. The annual evaporation is 2093 mm, annual average temperature is 6.2–8.7 °C, accumulative temperature $\geq 10$ °C is 2,900−3,500 °C, and yearly total radiation is 5.8 GJ/m$^2$ y. The soil type is dominated by loessial soil and accompanied by feldspathic sandstone landscapes, which are mainly chestnut soil and sand soil. This watershed is mainly planted with artificial species for soil-water conservation, wind prevention, and sand fixation. The major afforestation species include *Hippophae rhamnoides*, *Pinus tableulaeformis*, *Caragana korshinskii*, *Medicago sativa*, and *Prunus sibirica*. The major species under artificial *H. rhamnoides* forests include *Leymus chinensis*, *Stipa krylovii*, and *Cleistogenes squarrosa*.

### Experimental design
#### Methods

The decaying stumped *H. rhamnoides* artificial forest lands in the study area were selected in 2019 as the experimental field. Artificial *H. rhamnoides* forest lands with basically consistent site conditions, management measures, and growing conditions were selected as the sample plots. The basic information of the sample plots was listed in Table 1. The plots were all at the slope lower than 5°. In each plot the tree distance was 2 m × 4 m. The shrubs in the plots were stubbled differently in middle March 2019, including the stubble height of 0 cm (distance of 0 cm from ground, $S_1$), 10 cm ($S_2$), 15 cm ($S_3$), and 20 cm ($S_4$) as well as a control (no stubbling). Each stubble height was tested in triplicate. All sample plots were sized 30 m × 30 m. The standard plants in the sample plots were surveyed and marked by red paints. During stubbling, the plant samples sliced off were taken to the laboratory and oven-dried in an oven until reaching constant weights, which were measured. The heights of standard plants as-selected were measured in middle August of both 2019 and 2020. Thereby, the plant height compensatory growth of *H. rhamnoides* was calculated to analyze the two-year average values. At the same time, the ground biomass of each standard plant in each plot was monitored. Then, the average compensatory rate of ground biomass from August 2019 or August 2020 to March 2019 was calculated.

#### Detection of soil physiochemical properties

In August of each tested year, at horizontal distance of 10 cm from the standard plant in each forest land, a soil column at the 10 cm layers was collected using a soil auger, and used for analysis of soil physiochemical properties. The data were averaged between two years. For measurement of soil N, P the soil samples were wind-dried indoor, cleaned by removing fine roots and impurities, and passed a 0.25 mm soil screen. SOC was measured using the

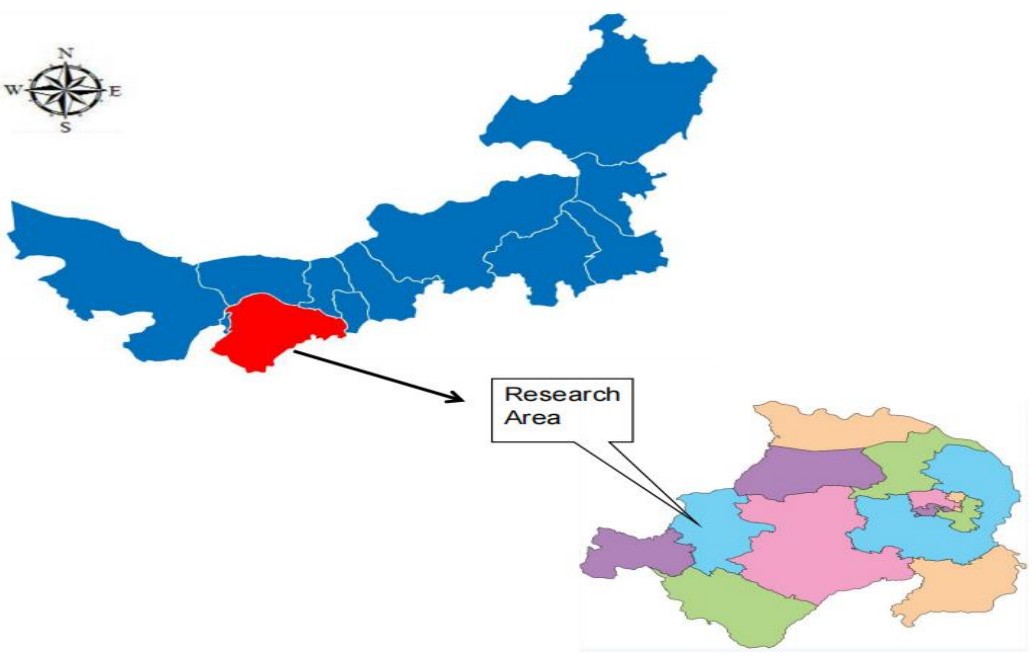

**Figure 1** **The location of the study area.** The study area is located in Geqiu groove watershed (39°42′–39°50′N, 110°25′–110°48′E) of Nuanshui Village, Jungar Banner, Ordos, Inner Mongolia. Figure source credit: 10.27229/d.cnki.gnmnu.2020.000695.

**Table 1** **Information of plots before stubbling.**

| Index | 0 cm stubble (S$_1$) | 10 cm stubble (S$_2$) | 15 cm stubble (S$_3$) | 20 cm stubble (S$_4$) | Control (CK) |
|---|---|---|---|---|---|
| Total coverage/% | 65.0 ± 1.15[b] | 58.9 ± 1.2[c] | 68.4 ± 0.51[a] | 64.6 ± 0.85[b] | 67.2 ± 0.67[ab] |
| Average height/cm | 98.63 ± 2.69[a] | 99.93 ± 3.07[a] | 99.13 ± 2.75[a] | 102.63 ± 3.25[a] | 100.87 ± 5.53[a] |
| Species richness | 5.2 ± 0.43[a] | 5.5 ± 0.41[a] | 5 ± 0.82[a] | 5.1 ± 0.7[a] | 5.2 ± 0.59[a] |
| Ground biomass/g m$^{-2}$ | 2536 ± 1.5[b] | 2673 ± 4.78[a] | 2635 ± 0.65[a] | 2673 ± 2.78[a] | 2622 ± 0.85[a] |

**Notes.**
Data are all expressed as mean ± standard deviation; different lowercase letters indicate significant differences among treatments ($P < 0.05$).

potassium dichromate external heating method. TP was detected using acid-dissolved Mo Sb colorimetric method. Soil pH was measured using the potential method. Soil moisture content was detected using the drying method. On a sunny day in August, soils were collected under each standard tree from the three repeated plots of each treatment and measured. Finally, the data were averaged. Total soil porosity was calculated as capillary porosity and non-capillary porosity according to the method by *Li et al. (2020)*. The soil capillary porosity and non-capillary porosity were detected using the ring cutter suction method (*Zhang et al., 2020*). On a sunny day in August, soils were collected under each standard tree from the three repeated plots of each treatment and measured.

## Compensatory growth indices

The compensatory growth mode was comprehensively judged according to the results of compensation indices and analysis of variance (ANOVA). Let GH be the plant height compensation, which is the sum of plant height harvested in August and plant height cut off in March; let GB be the ground biomass compensation, which is the sum of ground biomass harvested in August and ground biomass cut off in March. The plant height compensation index (GH/C) and the biomass compensation index (GB/C) can be calculated as follows:

GH/C = plant height at a specific stubble height/plant height of the control;

GB/C = ground biomass at a specific stubble height/ground biomass of the control;
Compensation index G/C >1 and significant difference between a stubble height treatment and the control; G/C = 1 and no significant difference; G/C <1 and significant difference indicate over-compensation, equal compensation, and under-compensation respectively (*Belsky et al., 1993*).

## Data processing and analysis

The differences in average height, ground biomass, soil physiochemical properties, and compensatory growth ability among different stubble heights were examined by one-factor ANOVA on IBM SPSS Statistics 24.0. Multiple comparisons were conducted using the least significant difference method (LSD). Relationship between compensatory growth ability of *H. rhamnoides* at different stubble heights and soil physiochemical properties was tested using redundancy analysis (RDA) on Canoco5. Plotting was finished on Origin 2019.

# RESULTS

## Compensatory growth ability of *H. rhamnoides* at different stubble heights

The values of GH/C under treatments $S_1$, $S_2$, $S_3$ and $S_4$ are 1.32, 1.35, 1.45 and 1.22 respectively, which are all larger than 1 (Fig. 2A). The values significantly increase by 31.50%, 34.75%, 45.25% and 21.75% respectively from that of the control (1.05 m). The GH/C is not significantly different among treatments $S_1$, $S_2$ and $S_3$ ($P > 0.05$), and the plant height compensation ability is the best after treatment $S_3$. The stubble height also affects the ground biomass of *H. rhamnoides* (Fig. 2B). The values of GB/C under different treatments are 1.13, 1.18, 1.24, and 1.15, respectively, which are all larger than 1. Compared with control (2.92 kg), the ground compensatory biomass at different stubble heights significantly rise by 12.75%, 18.25%, 23.75%, and 15.25% respectively. The GB/C is not significantly different among treatments $S_1$, $S_2$ and $S_4$ ($P > 0.05$). The biomass compensatory ability after treatment $S_3$ is the best (1.24) and is significantly higher compared with treatments $S_1$ (1.13), $S_2$ (1.18), and $S_4$ (1.15) ($P < 0.05$).

## Soil eco-stoichiometric variation characteristics of *H. rhamnoides* at different stubble heights

The physiochemical properties of understory soils in *H. rhamnoides* at different stubble heights changed in different trends (Fig. 3). In particular, the changes of TP and TN are synchronous, and rank both as $S_3>S_2>S_1>S_4>$Control. The changes of SOC rank as

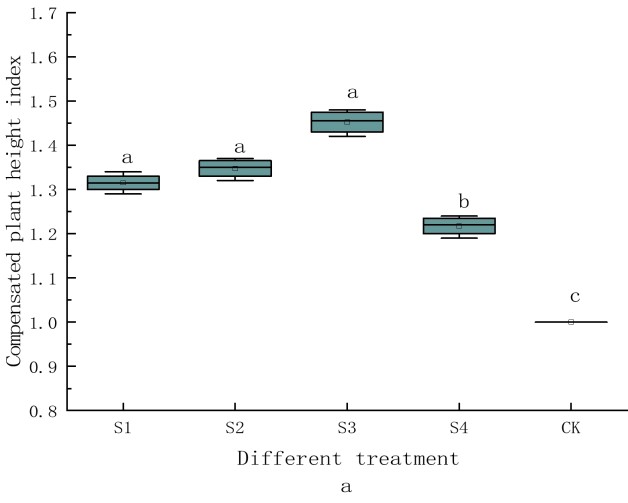

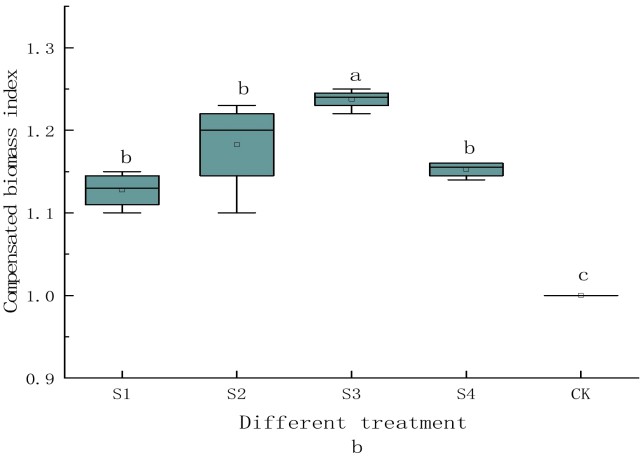

**Figure 2 Shoot height and biomass compensatory ability of *H. rhamnoides* at different stubble heights.** Different lowercase letters indicate significant difference between treatments.

$S_3 > S_4 > S_1 > S_2 >$ Control, and rise significantly by 40.62%, 58.97%, 44%, 36.96% from that of control ($P < 0.05$). The C:N ratios vary within 8.77 and 12.15 among stubble heights, and rise by 2.45%, 8.01%, 4.49%, and 0.42% respectively from that of control, but are not significantly different among the stubbling treatments. The C:P and N:P ratios both rank as Control$>S_1>S_4>S_2>S_3$, and the C:P and N:P ratios of $S_3$ decline significantly by 42% and 32.72% respectively compared with control ($P < 0.05$).

## Eco-stoichiometric correlations of understory soils in *H. rhamnoides* forests

The TP, TN, and SOC contents of understory soils are very significantly correlated ($p < 0.01$). The TP, TN, and SOC contents are all very significantly and negatively correlated with C:P (Table 2) ($p < 0.01$). The TP and SOC contents are both very significantly and

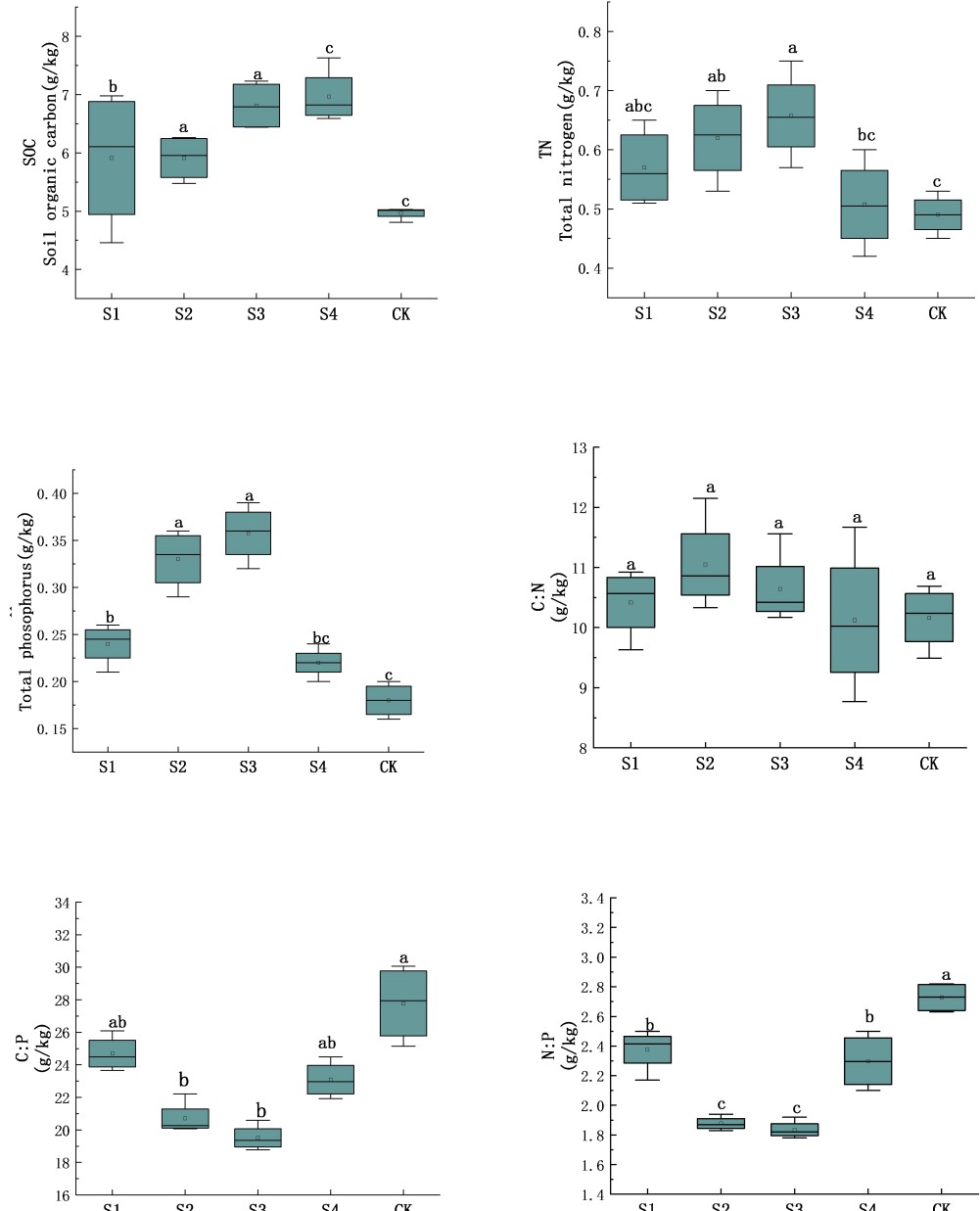

**Figure 3 Eco-stoichiometric characteristics in soil C, N, P contents of *H. rhamnoides* at different stubble heights.** Different lowercase letters indicate significant difference between treatments.

**Table 2 Variation coefficients of soil eco-stoichiometric ratios of *H. rhamnoides* at different stubble heights.** Eco-stoichiometric characteristics of soil C, N, P contents in *H. rhamnoides* at different stubble heights.

|  | TP | TN | SOC | C:N | C:P | N:P |
|---|---|---|---|---|---|---|
| TP | 1.0000 | | | | | |
| TN | 0.87** | 1.0000 | | | | |
| SOC | 0.96** | 0.89** | 1.0000 | | | |
| C:N | 0.1500 | −0.2700 | 0.2000 | 1.0000 | | |
| C:P | −0.91** | −0.73** | −0.78** | −0.0600 | 1.0000 | |
| N:P | −0.86** | −0.52* | −0.77** | −0.51* | 0.89** | 1.0000 |

**Notes.**

Pearson correlation analysis, two-tailed test; * significant correlation; $P < 0.05$, ** extreme significant correlations, $P < 0.01$.

*Correlation is significant at the 0.05 level.

**Correlation is significant at the 0.01 level.

negatively correlated with N:P ($p < 0.05$). N:P is correlated negatively with both TN, and C:N, and positively with C:P, all very significantly ($p < 0.01$).

## Effects of compensatory growth and soil properties on soil eco-stoichiometric characteristics of *H. rhamnoides* at different stubble heights

The eco-stoichiometric ratios of understory soils, compensatory growth under different stubble heights, and understory soil properties were sent to RDA. The explanation rates of axis 1 is 92.72% (Fig. 4), indicating the explained eco-stoichiometric ratio is correlated to different factors. SOC, TP, and TN are all positively correlated with total porosity, GH/C, soil moisture content, GB/C, water holding capacity, and stubble height, but are negatively correlated with soil bulk density and pH. N:P is correlated positively with soil bulk density and pH, and negatively with total porosity, GH/C, soil moisture content, GB/C, water-holding capacity, and stubble height. C:N, and C:P are correlated negatively with soil bulk density and pH, and positively with total porosity, GH/C, soil moisture content, and GB/C.

Soil moisture content can most largely explain the eco-stoichiometric feature of understory soils (81.6%), followed by total porosity (7.4%) (Table 3). The effects of soil moisture content and total porosity both are very significant ($p = 0.002$). The effects of soil bulk density, maximum water-holding capacity, stubble height, pH, GB/C, and GH/C on the eco-stoichiometric feature of understory soils are all less than 3.0% and insignificant ($P > 0.05$).

## DISCUSSION

The plant height and biomass compensation indices were both larger than 1 and over-compensatory growth occurred under any stubble height. The compensation growth indices differed among stubble heights, but were all significantly larger compared with the control (Fig. 2). First, the plant height compensatory growth was significant under stubble heights of 0 cm (1.32), 10 cm (1.35), and 15 cm (1.45), but was not significantly different among treatments. This may be because the ground heights of plant communities are stable
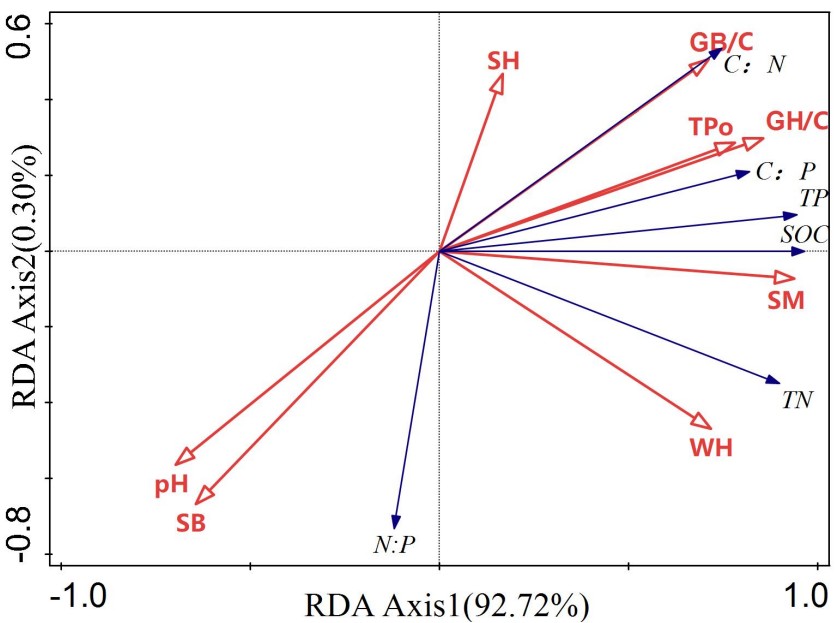

**Figure 4** **RDA of eco-stoichiometric ratio and soil properties.** SM, soil moisture content; SH, stubble height; SB, soil bulk density; TPo, total porosity; WH, water-holding capability; GH/C, plant height compensation index; GB/C, biomass compensation index.

**Table 3** **Eco-stoichiometric characteristics of soil C, N, P contents in *H. rhamnoides* at different stubble heights.** Importance sorting and significance test of explanation by environmental variables.

| Influence factor | Interpretive degree/% | Contribution/% | F | P | Sorting by importance |
|---|---|---|---|---|---|
| SM | 81.6 | 87.6 | 79.6 | 0.002 | 1 |
| TPo | 7.4 | 7.9 | 11.4 | 0.008 | 2 |
| WH (%) | 1.2 | 1.3 | 1.9 | 0.186 | 3 |
| SB | 1.5 | 1.6 | 2.6 | 0.146 | 4 |
| SH | 1.2 | 1.3 | 2.3 | 0.17 | 5 |
| PH | <0.1 | <0.1 | 0.2 | 0.704 | 6 |
| GB/C | 0.2 | 0.3 | 0.4 | 0.522 | 7 |
| GH/C | <0.1 | <0.1 | <0.1 | 0.966 | 8 |

**Notes.**

SM, soil moisture content; SH, stubble height; SB, soil bulk density; TPo, total porosity; WH, water-holding capability; GH/C, plant height compensation index; GB/C, biomass compensation index.

within a certain range of stubble height (0–15 cm), and the plant height compensation index under stubble height of 20 cm is lower than other three stubbling treatments. Second, during the revegetation of *H. rhamnoides* in feldspathic sandstone areas of Ordos, priority shall be given to the stubble height of 15 cm, which will create the best condition for vegetation recovery. The biomass compensation index under stubble height of 15 cm (1.24) is also significantly higher than the other three treatments. The possible reason for this result is that at the stubble heights of 20, 15, 10, and 0 cm, the upper, middle, and

middle lower parts of plants were cut off respectively. During the recovery of *H. rhamnoides* after stubbling treatments, the middle part of plants was under a relatively appropriate environment in the populations and was the fast-growing part. Hence, appropriate stubble height (15 or 10 cm) increased nitrogen distribution in the ground part (*Ilmarinen, Mikola & Vestberg, 2008*; *Sun et al., 2019*). This change offered resources for the reproduction of the ground part, and eliminated the apical dominance in the middle lower part and irritated to produce more new issues, thereby improving the tillering and photosynthesis of *H. rhamnoides* (*Du & Yang, 1989*) and accelerating growth and recovery (*Newingham, Callaway & Bassirirad, 2007*; *Zheng et al., 2017*). Thus, this may be one of the reasons why the biomass compensation of *H. rhamnoides* is the most obvious at the stubble height of 15 cm. All these variations led to over-compensatory growth after stumping. At the stubbling treatment of 0 cm, however, the whole ground part was cut off, and all ground branches and leaves were cleared away. Consequently, the reproduction of *H. rhamnoides* only depended on the nutrients stored in the basal part of stems, root collar, and roots. These nutrients played important roles in the initiation of reproduction and the early-stage growth of *H. rhamnoides* (*Meuriot et al., 2004*; *Dhont et al., 2003*). Compared with other stubble heights, the stubble height of 0 cm offered more space, mobile air and strong solar radiation for the growth of *H. rhamnoides*, which were favorable environmental conditions for the growth of new branches. Moreover, after whole-plant cutting, the upward nutrient transportation from the underground roots was shortened and quickened. Also, because the respiration and consumption by the old basal leaves disappeared, all nutrients in the roots can be utilized by new branches and leaves, which created a material condition for the growing leaves and branches. Because of the fast growth of new branches and leaves, the plant height compensatory growth was significantly higher than those of $S_4$ and control. The biomass of treatments $S_1$, $S_2$, and $S_4$ was significantly lower than that of $S_3$, but was significantly higher than that of control, indicating the stubble heights of 0, 10 and 20 cm affected the biomass similarly during the tested years, and the biomass under stubble height of 15 cm ($S_3$) was significantly higher than those of other treatments. This was probably because the removal of root meristems and aging tissues and the formation of lateral branches and new tissues (*Guo & Qi, 2018*) improved the illumination conditions in the middle lower parts and thereby increased net primary production. Under the stubble height of 0 cm ($S_1$), the whole ground part was removed, which decreased the photosynthesis ability and destroyed the growing points of plants, leading to an increase of death rate (*Hunt, 2001*). Besides, total removal of ground biomass in arid and semiarid areas caused the ground exposure, increased soil evaporation and hindered plant growth. Thus, the biomass compensation index under stubble height of 0 cm (1.13) was significantly lower than other stubbling treatments. Generally, appropriate stubble height ($S_3$ or $S_2$) can improve the biomass yield of communities, but low stubble height ($S_1$) may decrease the biomass yield. In all, the stubble height of 15 cm can well accelerate the growth of the ground part of *H. rhamnoides*.

Our results also show that the TP, TN, and SOC contents of understory soils in feldspathic sandstone areas of Ordos are 0.27, 0.57 and 5.95 g/kg respectively. Our results are close to the data in Jungar desert areas, but the TN concentration is higher compared with relevant data in Loess Plateau of China (0.40, 0.76, 7.77 g/kg) (*Zeng, Li & Dong, 2015*), the dry
farming areas of Northeast China (0.77, 1.43, 16.79 g/kg) (*Zhuo et al., 2019*), and Jungar desert areas of China (0.35, 0.21, 5.73 g/kg) (*Tao et al., 2016*). Our results indicate the nitrogen fixation ability in the roots of *H. rhamnoides* is significantly improved. According to Soil Nutrient Concentration Classification of China, TP is at very low level, and TN and SOC are at low levels. Basically, the soil nutrient concentrations in this feldspathic sandstone region are low and far lower than the average levels of China (0.65, 1.06, 11.12 g/kg) (*Tian et al., 2010*). Our results are consistent with the research by *Yang et al. (2005)*, suggesting the understory soil nutrients of feldspathic sandstone areas in Ordos are lean. The TP, TN, and SOC contents of understory soils under different stubble heights are all significantly different from those of the control ($P < 0.05$). *Snyder & Williams (2003)* stated that the soil organic matter contents (C and N) dropped after the removal of ground biomass following stubbling treatment, and such drop was less severe as the stubble height was shortened (*Hao et al., 2018*). Furthermore, stubbling treatment can trigger root growth, leading to an increase of SOC (*Ziter & MacDougall, 2013*). Our results show the TP, TN and SOC contents of understory soils after stubbling are all higher those of the control, which is consistent with the findings of *Ziter & MacDougall (2013)*. The soil nutrient concentrations under stubble height of 15 cm are significantly higher compared with other treatments (Fig. 3). One reason is ascribed to the roots grown in the *H. rhamnoides* under stubble height of 15 cm (*Liu et al., 2021*). The fixation by the huge root system can improve the erosion resistance of soils and restrict soil erosion, which increases soil silt and clay contents and optimizes soil properties. Another reason is that under the stubble height of 15 cm compared with the stubble heights of 0 and 10 cm, the biomass reserved in the middle and lower parts of each plant ensures the renewing and rejuvenation abilities, and the canopies can effectively block rain to decrease erosion from raindrop splash and decelerate understory potential evaporation, leading to an increase of soil temperature and humidity. This result is consistent with the study by *Han et al. (2014)* that stumping can improve nutrient contents and alter the relative distribution of nutrients, improving the nutrient-absorbing capacity of plants.

The average values of C:N, C:P, and N:P in understory soils are 10.48, 23.16 and 2.22 g/kg respectively, which are lower than the average levels of China (12.30, 52.63 and 4.20) (*Zeng, Li & Dong, 2015*). By comparing the C:N, C:P, N:P ratios of understory soils among different stubble heights, we think the above stable ratios may be explained by the long-term stable supply–demand relationship that exists between soils and plants during the growth of *H. rhamnoides* (*Gong et al., 2017*). The soil C:N ratio is not significantly different and is stable in soils (which means no severe variation within short time or at the space scale) (*Black et al., 2010*). Hence, the correlation analysis shows SOC is very significantly and positively correlated with TN. The C:P ratio is a characterization index of P availability, and a lower C:P ratio indicates higher soil P availability (*Yusup, Mansur & Nasima, 2015*). The C:P ratio in the study area is far lower than the average level of China, indicating the P availability of understory soils in *H. rhamnoides* forests is higher. *Yang, Chen & Zhang (2019)* also proved that at C:P < 200, the P release rate surpassed the P holding rate, thus increasing the P availability.

Redundancy analysis can uncover the coordinating relationships of soil physical factors with soil SOC, TN, TP contents and the eco-stoichiometric ratio, and help to more reasonably explain soil nutrients (*Li et al., 2021*). Soil moisture content is a key carrier of soil element migration and circulation and can directly affect soil nutrients and plant growth. Our results show soil moisture content is significantly correlated with the soil SOC, TN, TP contents and the eco-stoichiometric ratios of understory soils in *H. rhamnoides* forests ($P = 0.002$) (Table 3). *Feng et al. (2020)* found understory pores with excellent structures were well developed in *H. rhamnoides* forests, suggesting the increase of organic content is favorable for the formation of soil pores. Hence, soil moisture content and soil porosity are both closely related to soil SOC, TN, TP contents and their eco-stoichiometric ratios.

## CONCLUSIONS

This study was targeted at the decaying *H. rhamnoides* artificial forests in a feldspathic sandstone region. The plant height compensation ability and biomass compensation ability of *H. rhamnoides* at different stubble heights were studied. The relationships of plant height compensation ability, biomass compensation ability, stubble height and soil physicochemical properties with the eco-stoichiometric features of understory soils were further discussed. (1) Overcompensatory growth of *H. rhamnoides* in feldspathic sandstone areas occurred at all stubble heights. In particular, the plant height compensation index (1.45) and biomass compensation index (1.25) at the stubble height of 15 cm were both larger compared with other processing modes. Results show the stubble height of 15 cm can well promote the rapid growth of the ground part of *H. rhamnoides* in feldspathic sandstone areas of Ordos. (2) All stubble heights significantly affected the SOC, TN, TP contents and their eco-stoichiometric ratios in understory soils, but the changing rules differed. SOC, TN, and TP at all stubble heights were larger than those of the control, and maximized at the stubble height of 15 cm (6.97, 0.66,0.36 g/kg respectively). To maintain the ecosystem stability of feldspathic sandstone areas, the *H. rhamnoides* at the decaying phase shall be stubbled to the height of 15 cm, which will improve the soil physicochemical properties in these areas. (3) The understory C, N, P stoichiometric ratios of *H. rhamnoides* in feldspathic sandstone areas are jointly affected by compensatory growth, stubble height, and soil physicochemical properties. The understory soil eco-stoichiometric ratios of *H. rhamnoides* are mainly affected by soil moisture content (contribution of 87.6%) and total porosity (7.9%).

### Funding

This research was funded by the National Natural Science Foundation of China (31960329), the Autonomous Region Application Technology Research and Development Fund Program (2021GG0085 and 2019GG004), and the Inner Mongolia Ordos Application Research and Technology Development Project (2021YY-106-55). The funders had no

role in study design, data collection and analysis, decision to publish, or preparation of the manuscript.

## Grant Disclosures

The following grant information was disclosed by the authors:

National Natural Science Foundation of China: 31960329.

Autonomous Region Application Technology Research and Development Fund Program: 2021GG0085, 2019GG004.

Inner Mongolia Ordos Application Research and Technology Development Project: 2021YY-106-55.

## Competing Interests

The authors declare there are no competing interests.

## Author Contributions

- Xin Wang conceived and designed the experiments, performed the experiments, analyzed the data, prepared figures and/or tables, authored or reviewed drafts of the article, formulation or evolution of overarching research goals and aims, and approved the final draft.
- Yuefeng Guo conceived and designed the experiments, performed the experiments, analyzed the data, authored or reviewed drafts of the article, review and revision of the first draft, and approved the final draft.
- Wei Qi conceived and designed the experiments, performed the experiments, authored or reviewed drafts of the article, review and revision of the first draft, and approved the final draft.
- Li Zhen conceived and designed the experiments, performed the experiments, analyzed the data, prepared figures and/or tables, review and revision of the first draft, and approved the final draft.
- Yunfeng Yao conceived and designed the experiments, performed the experiments, authored or reviewed drafts of the article, verification of experimental design, and approved the final draft.
- Fucang Qin conceived and designed the experiments, performed the experiments, authored or reviewed drafts of the article, verification of experimental design, and approved the final draft.

## Data Availability

The raw measurements are available in the Supplementary File.

## Supplemental Information

Supplemental information for this article can be found online at http://dx.doi.org/10.7717/peerj.13363#supplemental-information.

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
