# Peer review of "Compensatory growth and understory soil stoichiometric features of Hippophae rhamnoides at different stubble heights"

_PeerJ, doi:10.7717/peerj.13363_

## Round 0.1 · original submission · Major Revisions

We have received two reviews on your manuscript. I agree with the reviewers that the topic is relevant to the journal, the methods are appropriate, and the findings are interesting and valid in general. However, the manuscript must be improved according to the reviewers' comments. In addition, the English needs to be improved throughout the manuscript.

Reviewer 1 ·

Basic reporting

The effect of stubble heights on height growth and biomass of Hippophae rhamnoides, which is crucial and useful species used in the vegetation recovery practice in that area, were reported in the paper, it is interesting to provide the appropriate stubble height to help local people to manage this vegetation in the fragile environment by two-year experiment. I recommended it publish after minor revision.

Abstract section, in the part of Method, “The findings will offer some new clues
for eco-construction in feldspathic sandstone areas and theoretically underlie soil-water
loss administration in this region” this sentence should be moved to the end of Result part.
Line, 90, changed “Yu Wentao” to “Yu”
In the Introduction section, a hypothesis is needed.
Line 131, change “vegetation” to “species”.
Line 222, in this section, “The explanation rates of axes 1 and 2 are 92.72% and 0.30% respectively”, because the explanation rate of axis 1 was 92.72%, and it is high enough, and it was only 0.30% for axis 2, it is better to ignore the axis 2.
In the result section, please provided the statistically significant level, such as P<0.05 or P<0.01.
Line 239-241, please decide the compensatory type based on the statistical analysis results, although, the index was larger than that control, it should be statistical larger, then you can get such conclusion.
Line 280-281, the statement was different from the results, please check.
Line 295, check the authors.
Line 328-330, why did you cite this result, RDA, not PCA was used in this research.
Line, 345, “1.48” was different from the data in the result section.
All Figures need improve to make them clear. Please indicates the full name for abbreviations in the Fig. 4.

Experimental design

The experiment was well designed.

Validity of the findings

The findings were validated by data and statistical analysis.

Reviewer 2 ·

Basic reporting

1. The manuscript entitled “Compensatory growth and understory soil stoichiometric features of Hippophae rhamnoides at different stubble heights” investigated soil carbon, nitrogen, and phosphorus and their stoichiometric characteristics, as well as the compensatory growth ability of Hippophae rhamnoides at different stubble heights. The results find that the stubble height of 15 cm promoted the growth of the vegetation and the accumulation of soil carbon, nitrogen, and phosphorus. Overall, there are some interesting findings. However, I have some concerns about the logic, arguments, and wording in the paper. For example, the level of English throughout the manuscript does not meet the journal's desired standard. Please check the manuscript and refine the language carefully.

2. The “Introduction” section needs to be expanded further with relevant information. For example, why do the findings differ between the existing studies and what are the possible reasons? The authors may summaries appropriately, thus drawing out the importance of your research, rather than simply highlight the need for further research because of the inconsistent findings among previously studies.

Experimental design

1. Line 224-232: You have used variables such as total porosity and water holding capacity in RDA, but how did you measure these variables? All the variables used in your study should mentioned in Materials & Methods.

Validity of the findings

1. Line 205-207: What is the purpose of analyzing the variation coefficient of the variables within each treatment? A weak variation of coefficient for each variable among replicates within the same treatment is normal; in contrast, a large variation of coefficient indicates high heterogeneity between samples or a problem with the experimental design.

2. Line 233: Soil moisture is a highly dynamic variable and is sensitive to weather etc. Are the results of soil moisture reliable if only measured once? Please clarify.

3. Line 234-235: “The effect of soil moisture on total porosity is very significant.” How does soil moisture affect soil total porosity? It should be the other way around.

4. Generally, the “Discussion” is mainly descriptive, similar to the section of “Results”. There is little explanation of the indicative meaning behind the observed results. Authors should compare the findings to previous studies. Also, the deep meaning of the results should be described from a larger perspective.

5. The conclusion is also descriptive and lengthy. In the conclusion section, authors should focus on one or several important conclusions based on your results and highlight the importance of your study or state what gaps your research fills.

Additional comments

1. Line 23-24: “Method. The findings will offer some new clues for eco-construction in feldspathic sandstone areas and theoretically underlie soil-water loss administration in this region.” This sentence looks more like a research objective and cannot be placed in the Methods section of the Abstract.

2. Line 27: “Investigated by RDA”? How should this statement be interpreted? RDA is a method of analysis, how to investigate the compensatory growth ability.

3. Line 36: Abbreviations should be given their full name when they first appear in the abstract.

4. Line 38: “SOC, TN, TP contents (0.36, 0.66, 6.97 g/kg)”. It doesn't make sense that the content of TN is nearly half as high as SOC. Please check the experimental analysis process or data processing methods.

5. Line 41-43: “SOC, TN, and TP are all dominant factors to regulate the eco-stoichiometric features of understory soils in H. rhamnoides”. How can SOC, TN and TP be the dominant factors in regulating eco-stoichiometric characteristics when they are themselves indicators for calculating eco-stoichiometric characteristics in your study? This is difficult for me to understand.

6. Line 74: Please add the year of this reference after Cutforth et al.

7. Line 78: Also add the year of this reference after Cutforth et al.

8. Line 84: Add the year of this reference after Shahrudin et al. It is the same elsewhere.

9. Line 154-156: Abbreviations should be given in full when they first appear in the paper.

10. Line 158-165: The method for determining soil pH is very general and does not have to be so detailed (8 lines in the paper), please refine it.

11. Line 173-181: Please add the related references.

12. Line 187: What method is used for multiple comparisons after ANOVA?

13. Table 1: Note at the end of the table that the data in the table are means ± standard deviation or standard error. In addition, a test for significance of differences is required.

14. Table 4: Abbreviations should be given in full at the end of the table.

15. Fig. 4: Please standardize the names of the variables in the diagram, e.g., PH, SoilBulDn, etc. In addition, abbreviations should be given in full in the note.

---

## Round 0.2 · Minor Revisions

Please make the changes as suggested by the reviewer.

Reviewer 2 ·

Basic reporting

There are many punctuation issues throughout the manuscript. Do not put spaces between the punctuation mark and the preceding words. Spaces are used to separate words, and punctuation should follow the last word immediately. For example, Line 38: “of 15 cm .”, it should be “of 15 cm.”.

Additionally, there must be a space between each sentence and the punctuation of the previous sentence. For axample, Line 57: “erosion.Hence,”, should be “erosion. Hence”.

There are many such errors throughout the manuscript, check the full manuscript carefully.

Experimental design

No comment

Validity of the findings

No comment

Additional comments

1. Line 35: it should be “soil organic carbon (SOC)”, not “soil rganic arbon(SOC)”. Many similar problems, correct them seriously.
2. Line 35 and 38: it should be “nitrogen”, not “itrogen”. “Word” has the function to check for misspellings, why don't you check carefully before submitting your manuscript?
3. Line 37: “at all stubble heights were larger than those of the control”. The abbreviation CK for control was used earlier, so why not continue with the abbreviation here. But in fact, CK is not the correct abbreviation for control. Control and CK are both one word, so why must you use abbreviation?
4. Line 93: “potential of yhrogen”???
5. Line 165: Replace “+” with “and”.
6. Line 171: If you have removed the coefficient of variation from the “Results”, why do you need to write the formula for calculating the coefficient of variation in the “Methods”?

---

## Round 0.3 · accepted · Accept

Thank you for revising your manuscript, which now looks much improved.